# The role of doulas in abortion care in Canada: A qualitative study

**Martha Paynter**[ID][1], **Clare Heggie**[2], **Anja McLeod**[ID][3], **Mélina Castonguay**[4], **Melissa Fuller**[5], **Wendy V. Norman**[ID][6]

1 Faculty of Nursing, University of New Brunswick, Fredericton, New Brunswick, Canada, 2 Department of Interdisciplinary Studies, University of New Brunswick, Fredericton, New Brunswick, Canada, 3 Department of Sociology, Dalhousie University, Halifax, Nova Scotia, Canada, 4 Independent Consultant, Quebec, Canada, 5 Action Canada for Sexual Health and Rights, Ottawa, Canada, 6 University of British Columbia Faculty of Medicine, Vancouver, British Columbia, Canada

## Abstract

### Introduction

Equitable and safe access to abortion is essential for reproductive autonomy. Despite decriminalization in 1988, barriers to accessing abortion in Canada persist, particularly for people in underserved groups. Doulas, working in a non-clinical and unregulated supportive role, may facilitate access to family planning services, but evidence about the scope, role, training, and efficacy of doulas in abortion care is lacking. The goal of this study was to understand the experience of abortion doulas in Canada with respect to their role and facilitators and challenges faced in practice.

### Method

For this community-based, qualitative national study of abortion doula practice, we first created a national directory of all doulas in Canada in 2023 using publicly available online information. From this directory, we extracted contact information for doulas who indicated they provide abortion support and sent invitations for a one-time qualitative interview. Questions addressed current role and responsibilities, gaps in doula training and knowledge, and challenges and opportunities they perceived with respect to client support. We used reflexive thematic analysis to generate key themes. Our research team includes abortion doula experts.

### Results

Among 73 abortion doulas with public listings in Canada in 2023 to whom we sent invitations, 15 completed an interview. We developed six key themes: *Inter and intra-systems navigation; After-care; In over their heads; Balancing act; Support for the supporter; and Beyond the individual.* Abortion doulas navigate fragmented systems and support clients experiencing intersecting forms of oppression and exclusion. These complex logistics include locating care providers, securing transportation, and obtaining funds for clients' out-of-pocket costs. Doulas faced challenges acquiring adequate training, understanding

**Data availability statement:** Data cannot be shared publicly due to ethical obligations to protect the anonymity of participants. Data are available from Dr. Patricia Morris (contact via p.morris@unb.ca) for researchers who meet the criteria for access to confidential data.

**Funding:** Funding for this study was provided by Health Canada. The funders did not play any role in the study design, data collection and analysis, decision to publish, or preparation of the manuscript.

**Competing interests:** The authors have declared that no competing interests exist.

local health systems, forging relationships and supporting clients facing structural barriers to care. Abortion doula work is facilitated by peer and organizational support, connections to clinics and providers, and comprehensive and contextually appropriate training.

## Conclusions

Abortion doulas express needs for comprehensive, evidence-based training, meaningful partnerships and sustainable remuneration. Future research should explore abortion provider and clinic staff perspectives with respect to the potential role of doulas in wraparound client support services.

## Introduction

### Abortion in Canada

Equitable access to abortion is essential to promoting and protecting reproductive autonomy. Medication and procedural abortion are safe, common and affirmed globally as essential healthcare [1]. In Canada, abortion was completely decriminalized in 1988 [2]. Abortion is publicly funded with few exceptions, and self-referral is available in all provinces and territories [3]. The vast geography of Canada and small number of procedural abortion providers concentrated in urban centres long presented serious travel burdens. Access substantially expanded in 2017 with the first availability of mifepristone for medication abortion, with further improvements over the next two years as governments universally subsidized the medication, approved dispensing directly by community pharmacists, and authorized prescription by primary care physicians and nurse practitioners [4,5]. The availability of mifepristone led to significant growth in the number of abortion providers in Canada, and a rapid increase in the proportion of medication abortions amongst all abortions [6–8]. In recent years, the growth of telemedicine has further improved access pathways [9,10].

### Barriers to care

Nonetheless, numerous logistical barriers to abortion care persist in Canada that may be particularly complex to navigate for people in underserved groups such as youth, people experiencing housing insecurity and poverty, members of two-spirit, lesbian, gay, bisexual, transgender, or queer (2SLGTBQIA+) communities, people with disabilities and Indigenous people and people of colour. Youth experience confusion about their rights to seek care without parental consent or involvement. Medication abortion to induce a miscarriage at home is inappropriate for patients without a safe home. Although there are no legal restrictions on gestational duration, limits exist based on facility policy and provider preference and training and availability of abortion care in the second and third trimesters is limited [11]. Those who may need this care the most, who face barriers to accessing abortion in the first trimester, may likewise face considerable hurdles to travel to the few locations where later care is offered. Although abortion is publicly insured under Medicare, people living in Canada without a Medicare card, such as undocumented immigrants, temporary foreign workers, and international students may be required to pay out of pocket or seek assistance from community-based and charitable organizations [12,13]. These constitute a small but important number of people, estimated at 0.1% of patients [7]. Patients may also incur private costs such as child-care, gas and accommodations, and missed time at work; these costs could be higher where greater travel to care is required [14].

Access-limiting policies at both provincial and clinic/hospital levels may be informed by pervasive and historical abortion stigma [15]. Researchers have used Goffman's concept of stigma as an "attribute that is deeply discrediting" [16] to understand how abortion stigma can create barriers to accessing care. Abortion stigma can prevent patient's from seeking care due to fear of discrimination and perpetuate harmful misconceptions about abortion safety and access pathways [17–19]. Some patients may be at a greater risk of experiencing stigma from care providers, such as patients who use substances and patients working in the sex trade may encounter discrimination [20, 21].

Furthermore, there are aspects of care delivery that present barriers. Some clinics conduct all communication by phone, which patients living without cell phones or using text-only plans are likely to find challenging. Clinic requirements that patients have a drive home after procedural abortion may be a serious barrier for patients with few social supports [22]. Health facility environments may be frightening for people who have experienced individual and intergenerational trauma in hospital settings, particularly in relation to reproductive care. For example, Indigenous people in Canada have disproportionately experienced forced and coerced sterilization and the "birth alert" practice of newborn removal by child protection services [23]. Heteronormativity in reproductive care approaches and language excludes queer patients, making care unapproachable [24]. People with disabilities may face inaccessible clinic entrances, uncomfortable exam tables, and other physical interference with access [25].

Intersecting experiences of oppression threaten agency of abortion-seekers, and inhibit their access to information about care, potentially resulting in delayed or denied service. Doula support could play a role in mitigating these issues.

## Abortion doulas

While definitions vary, a doula is typically defined as a non-clinical support person who provides emotional, physical, and informational support throughout the perinatal period [26]. The positive impact of doula support on birth and postpartum outcomes is well-researched, with reported reductions in peripartum mood disorders, reduced operative deliveries, and increases in breastfeeding initiation and birth weights [27–29]. In the context of increasing recognition of high infant and maternal mortality among Black patients in the US, researchers have described birth doulas as especially beneficial for clients who face racism, poverty, and gender discrimination from perinatal health care professionals [30–34]. Given evidence birth doulas improve patient outcomes and reported satisfaction [35], there is increasing interest in better understanding the role doulas can play in abortion support and care [36]. Doulas supporting abortion are sometimes referred to as "full-spectrum" doulas, indicative of support for all pregnancy outcomes, including birth, abortion, miscarriage and stillbirth [37]. Some doulas may solely provide abortion support services. Like birth doulas, abortion doulas could be especially beneficial for clients who face structural barriers to accessing appropriate care, including discrimination from family planning health care professionals. Researchers have described inequities in abortion care access and experiences for Black [38, 39], Indigenous [40, 41], Latinx [42]; and 2SLGBTIQA+ patients [43].

Abortion doula training is heterogenous. As with perinatal doulas, there is no one regulatory or certifying body, and no required curriculum. Some of the largest and long-established doula training organizations do not offer any content on support for abortion-seekers [44]. However, some doula organizations do offer modules or continuing education on abortion [45–47]. There are organizations offering stand-alone, comprehensive abortion doula trainings that do not require perinatal doula experience, including community organizations and/or volunteer run abortion support networks [48–51].

Existing research, while limited, indicates that clients with abortion doula support report high patient satisfaction and positive experiences [52–54]. Our 2023 international scoping review of the role of doulas in abortion care identified 33 studies, the majority located in settings where abortion is highly restricted in Asia and Africa [55]. The review found abortion doulas support both procedural and medication abortion, assist clients in navigating complexities in healthcare systems, and support clients facing legal and political barriers [55]. The review identified no studies in Canada. Our subsequent research found approximately 10% of doulas in Canada support clients seeking abortion [56]. Our objective in this project was to understand the experience of abortion doulas in Canada with respect to their role and their training and mentorship needs.

## Methods

### Theoretical framework

We used a theoretical framework of reproductive justice to guide the study design and implementation. Reproductive justice theory was conceptualized in 1994 by twelve Black women working within the reproductive rights movement [57, 58]. Seeking to better understand the role of doulas in abortion care within a broader framework of reproductive justice, we maintained an expansive understanding of what actions might constitute the work of an abortion doula and sought to understand how abortion doulas own identities may influence both the understanding of their own role and their perceptions of client experiences.

### Design

We used a community-based approach to this qualitative study. In community-based research, community partners are engaged at each step of the research process to validate and provide insight into key themes and to support translation of knowledge into practice [59]. Our team consisted of both academic researchers and non-academic abortion doula experts who are situated in the community and have extensive experience with abortion doula practice and training. While some academic team members had experience providing abortion doula services, this partnership did not pre-exist prior to the research project. It was developed with the aim of conducting collaborative research on abortion doulas in Canada. Academic team members specifically reached out to two abortion doulas who were recommended by community partners as experts in their field both regionally and nationally, who represent both anglophone and francophone communities in Canada. They accepted invitations to join the research team as content experts and collaborators and were compensated with a stipend for their time on the project. Academic and non-academic partners worked collaboratively and iteratively through regular meetings and consultation. Study team members included people with experience as abortion doulas and providers, and people who identify as queer.

We used purposive sampling. To identify abortion doulas to invite for interviews, we first created a directory of all the doulas in Canada we could identify with publicly available online information, including websites, social media and listings on doula association websites [56]. We then extracted all doulas in the directory who explicitly stated that they provide abortion support in their online materials. A research assistant sent a recruitment email to each of the doulas identified as providing abortion support on July 5th, 2023. The recruitment email included details on the study aims, interview procedure, and participant confidentiality. Abortion doula experts provided feedback on the email script and were also invited to share the script with abortion doulas in their networks. Only doulas with an email contact on their online materials were eligible. If interested, participants responded to the research assistant to schedule a zoom interview on a day and time that was convenient to them. We used snowball

sampling among interview participants to expand recruitment by encouraging them to share study information with other abortion doulas in their networks. We closed recruitment on October 23rd, 2023. Participants had the option to conduct the interview in English or French. All participants received a $50 gift card honorarium for their participation. Audio-recordings were de-identified and transcribed by a professional transcriptionist. This study was approved by the University of New Brunswick Research Ethics Board.

## Procedures

Interviews were guided by a semi-structured interview guide developed in consultation with the abortion doula expert team members. We asked about participants' demographic information; experiences as a doula; years of practice; frequency and/or volume of clients; and characteristics of clients (e.g., procedural vs. medication, gestational duration limitations or norms); how they received payment for services; what training they had received and what the curriculum included; the challenges and facilitators they experienced providing doula services; what supports abortion doulas need; and issues specific to their region/jurisdiction. One research team member was present on the call with the participants to conduct the interview. Abortion doula expert team members did not conduct interviews.

## Analysis

We used a collaborative process of reflexive thematic analysis to analyze de-identified transcripts. First, research team members reviewed the raw data to gain familiarity with emerging themes. Two research team members double coded all interviews and iteratively developed a coding scheme that was adapted through discussion between coders and consultation with the entire team, while reflecting on what our identities and experiences bring to the analysis. The entire team then reviewed codes to develop broader themes. Finally, themes were interpreted and data was synthesized. To establish trustworthiness in our approach, abortion doula expert team members reviewed and edited the organization of themes over the course of several meetings. This collaborative process supports the dependability and credibility of our findings.

## Results

Two team members conducted data collection (CH and MP) in summer and fall 2023. Of the 73 abortion doulas identified in the national directory, 15 participated in interviews. Participants shared a range of experience as abortion doulas, with some new to practice and some having practiced for over a decade. Participants lived in five provinces and one territory, with four participants each from Ontario and Quebec, three from Alberta, two from Nova Scotia, and one each from the Northwest Territories and British Columbia. Two participants identified as Indigenous. Two participants identified as Queer. Three participants identified as Francophone; one participant conducted their interview in French with a bilingual research team member. Three participants identified as living and practicing in a remote or rural community. The interviews were, on average, approximately 45 minutes in length.

We identified six key themes through iterative and collaborative analysis: Inter and intra-systems navigation; After-care; In over their heads; Balancing act; Support for the supporter; and Beyond the individual. The six key themes speak to the work of abortion doulas at both micro (individual service relationships) and macro (community/society) levels, including the ways abortion doulas challenge and collaborate with the formal healthcare system, the complexity and precarity of abortion doula practice, and the potential for abortion doula work to contribute to visions of reproductive justice.

### Theme 1: Inter and intra-systems navigation

Abortion doulas worked both inside and outside of the formal health system, assisting clients to navigate logistical barriers such as transportation to appointments, funds needed for travel or supplies such as menstrual pads and over-the-counter medications (e.g., acetaminophen, ibuprofen), and lost income from missing work. Doulas characterized these barriers as exacerbated in rural or remote areas. As one doula working in a rural area described:

> *The lack of access to the services is an issue but also the lack of financial resources for the people who do have to travel, have to take time off work, have to pay for gas, or have child-care, what have you, all of those things are an issue specific to any area outside of the city. (Participant 9).*

Abortion doulas working in urban areas also supported rural clients who traveled to the city, meeting them at airports or train stations, providing gas cards, arranging hotels and even helping youth and/or people without credit cards to check-in to hotels. They offered support to clients unfamiliar with the city environment:

> *Most of the support we provide is to people coming from out of town, which makes sense, there's more options here, but it's definitely a challenging situation because it requires a lot more logistical coordination, like we're asked by partners to do a lot more of kind of, I mean I think this is also part of doula work, but a lot more kind of grunt work. Help somebody go and check someone into a hotel or go and buy pads for the person. So sometimes the extent of our doula support is us showing up, buying the thing, and the person, they don't want doula support, and that's okay (Participant 10).*

Participants consistently described transportation as a key aspect of abortion doula work. Several doulas expressed frustration with the inadequate provincial/territorial compensation for patient travel to appointments. For example, mileage reimbursements capped at 250 km, or max accommodation rates that were far too low for what is required in the city. One urban abortion doula explained their doula network spends the bulk of their time sourcing funds for client travel, including paying out of their own pocket:

> *Right now we're paying [taxes] for a healthcare system, and then we're paying to send people to healthcare. (Participant 12).*

Participants explained how they support clients by steering them through and between complex health and social service systems. Some stated clients often reached out before booking anything because they were uncertain about "where to start". This was particularly common for clients from rural and remote areas and for people seeking care later in pregnancy, when travel for care is more likely to be required. Much of the information doulas describing themselves as frequently sharing with clients was specific to their location, for instance, local gestational duration capacity, and what procedures and pain management options are available from a given care provider or facility.

### Theme 2: After-care

Abortion doulas recounted providing their clients with continuous emotional and physical support. They explained their clients perceived a lack of follow-up or after-care available from the formal (publicly funded) health system during and after abortion. The participants explained their clients felt support from healthcare providers (HCPs) often stopped abruptly

after the procedure or prescription for mifepristone. They acknowledged that while many clients do not want or need follow-up support, others continue to reach out for months after the abortion:

> Some of our clients it really is just that they need to get from point A to point B and you're just accompanying them. But with some of our clients, I've worked with some clients for up to six months, where it's not that we're speaking every day but maybe once a month they reach out cause they want to talk and you're the only person in the world who knows they had an abortion. (Participant 10).

Emotional support took many forms. The abortion doulas described being available for phone calls and texting with clients after a procedural abortion or while clients experienced medication abortion at home, making referrals to other clinical or social resources as needed. Indigenous doulas described the culturally relevant emotional support they provided to clients in their own community. All participants stressed the importance of following the client's lead and being prepared to support any emotional response on the part of the client:

> We don't induce the idea that the woman will have some grieving, we listen and we offer help. We can offer ideas for ritualizing, if the woman needs it, but it's very important that we don't induce that she should need it, that she should be sad. We are open to what this woman is precisely going through, so it can be a liberating moment too. (Participant 4)

The physical support role of the abortion doula varied depending on whether the client experienced procedural or medication abortion, and if the facility had a policy in support of or restricting the presence of companions. For staff and patient security and privacy, many clinics do not allow any support people in the clinics; some allow support people to wait in a separate area. Physical and emotional support overlap as abortion doulas may accompany the client to the appointment and then wait in the car or waiting room due to client preference or clinic policies, to be available for immediate post procedure support.

## Theme 3: In over their heads

Abortion doulas described the complexity of their work. Some, who had previously supported births, expressed surprise at the needs of abortion patients, stating that at certain points they felt "in over their heads". Acknowledging not all abortion access experiences are complex, participants explained that clients seeking abortion doula support were often those facing intersecting barriers and special considerations necessitating extra support, such as clients living in rural and remote places, experiencing intimate partner violence, houselessness or substance use. They believed their training did not adequately prepare them for these situations:

> I think the major gap for doula training is that we don't get specific training for those individual populations. So you might feel most comfortable dealing with somebody who is maybe in their 20s or 30s but dealing with youth, dealing with situations that involve sexual assault, incest, things like that is I think just a discomfort for people, they don't know how to interact with folks like that, they don't know the situation, they don't know how much to ask or not ask. People who are sex workers, people who use substances. I think those are groups where people have a really difficult time navigating those situations. (Participant 14)

Participants also described their training as having left them unprepared for clinically complex cases, such as procedural abortion in the second or third trimester. Care in these cases

may include a fetal injection to terminate the pregnancy, followed by the induction of labour, and can involve several types of HCPs. While abortion care in the first trimester is often provided by family physicians and nurse practitioners in primary care clinics, abortion care after the first trimester is provided in hospitals by maternal-fetal medicine specialists, obstetricians and midwives.

*The first time we were asked to support an induction, it was completely like we were figuring it out as we went, there were no resources, we didn't really know how to show up. I mean I was loss trained, most of our abortion doulas had pregnancy loss training because in the absence of abortion training that's a lot of what we would do and sometimes folks are accessing abortions for a wanted pregnancy, so there's the overlap but it still just wasn't quite the same. (Participant 10).*

Abortion doulas believed training should include comprehensive information on "what actually happens" clinically to adequately prepare doulas for practice and support. They felt that while perinatal doula training should cover all pregnancy outcomes, people wanting to provide abortion support should seek out additional resources and information and be self-aware about their capacity to support without judgment in second and third trimester cases. Participants most satisfied with their training reported that they took a stand-alone, comprehensive abortion doula training. Some were skeptical of brief (e.g., two hour or half day) training offerings, expressing how complex the work can be. Participants acknowledged the importance of staying within the doula's non-clinical scope, but felt that doulas needed to understand what happens clinically to properly answer client questions, including knowing when to direct them back to their provider:

*If the client calls you and they're like I'm having a D&C on Monday, you don't want to be asking the client what that is. If you know what that means, you have more of a sense of what they're going to be going through on that day. Even if they were to ask specific questions and you're kind of like, it would be really great to ask your provider, or they'd be the best authority on it because it can look a little different depending on what clinic you're at or whatnot, it's still so important to have that information. (Participant 10).*

Doulas practicing in rural areas described feelings of isolation and inadequate preparation from trainings that are typically offered in urban areas and that refer largely to formal health system services available in urban areas. Rural participants expressed how developing knowledge of regional-specific resources was an important facilitator for their practice. One abortion doula working in a rural area described their training experience:

*So many trainings are not in [region] so that would be really, really ideal for people here. When I did my doula training it was very much based, obviously we were in [urban centre], and so many times I would be like, oh guys we don't have those services and we don't have those technologies here yet or there are all of these things, I was like this isn't applicable to our region. (Participant 9).*

### Theme 4: Balancing act

Participants recounted maintaining a careful balancing act between their abortion doula practice, advocacy, other professional commitments, and personal lives. Further, participants described an ethical balancing act between their desire to provide abortion support at little to

no charge to clients in need of support and their need to sustain themselves financially. Many provided services on a volunteer-basis, either in addition to a birth doula business, or on the side of unrelated full-time work. Participants described experiencing secondary or vicarious trauma and burnout, with burnout sometimes resulting in participants taking a step back from abortion doula work either temporarily or permanently. Participants expressed feeling awkward and ethically uncomfortable asking for money to support an abortion, and had difficulty aligning their personal needs to make a living with a reproductive justice or equity-based lens to doula support:

> I can't expect that someone is able to pay me and I don't want to ask for payment all the time as well, but I also need to pay my own bills so it's a difficult kind of balancing act. (Participant 2).

Some described working within a group as one approach to balancing sustainable coverage and funding with the goal of providing low-barrier support. In this model, the doulas received compensation from a community-based organization, and clients received support at no cost. This model presents its own challenges, as community-based organizational funding may be tenuous and dependent on government funding or private donations and may be reliant on pro-choice sentiment. Doulas expressed frustration that a lack of continuous and long-term funding for abortion support services negatively impacted both clients and other community organizations:

> So [organization], it may close this year because of lack of funding so if there are no resources they may not be here a year from now. Our community partners, they expect us to help them, and they're also depending on funds, so for them it's, oh another resource that we're going to lose. (Participant 13).

Other participants belonged to volunteer networks. While this model helped with distribution of client load, and members chose to forego compensation, they expressed the belief they were filling gaps in service that could or should be addressed by the formal health and social systems. Many doulas described paying for things like gas, hotels, and groceries for clients out of their own pocket. In both community-based organization and volunteer network models of abortion doula work, the onus was placed on doulas and other community partners to fund and sustain their practice. Some participants felt these models were short-term or 'band aid' fixes for the root problem: poverty leads to unequal access to health care, despite theoretically universal coverage for people in Canada. As one doula summed up when describing fundraising for a client's travel needs:

> I mean there's only so much you can do as an individual or as a community group, you're just putting Band-Aids on things. This is something that is not going to get any better without the government just making a commitment to the wellbeing of its citizens. (Participant 12).

Another possible model of abortion doula work described by participants was integrating doulas directly into abortion clinics, as employees within the formal health system. While some participants felt this would be a "best-case scenario", others valued their independence and status as unaffiliated with the formal health system and felt this would force doulas to compromise between client needs and responsibility to the employer. This participant described how client preferences determine their level of partnership with the formal health system:

*I've supported some clients in filing complaints against their providers. How comfortable would I feel doing that if I was working within a clinic that I have a relationship with? I think it can get tricky in terms of being able to prioritize your client. I think every doula who has ever been in a hospital setting at a birth knows how weird interactions get with staff and how hard it can be to name when something is happening that is problematic, that might be violent or abusive in some way, it can actually be extremely hard because there are all these like invisible dynamics that pop up that you weren't even aware of outside of it. And so I think the relationships we hold in the spaces are really important and I always think the client is the most important one. (Participant 10).*

## Theme 5: Support for the supporter

Participants described strong community ties to peers, organizations, and abortion providers as an important facilitator of their work. Members of abortion doulas networks described feeling supported and confident in their work, protected from burnout and secondary trauma. Peer support helped them to navigate complex client cases and the intricacies of the work, such as if or how to ask for payment and whether and how to apply for insurance. In some cases, being part of a community allowed doulas to access compensation via organizational grants or donations. One participant described the impact of peer support on their work:

*I'm seeing way less of a gap there in the resources and connection than when I started, and so I guess I'm sort of living in this beautiful dream world of it being so much community building and people are so supported especially on that administrative side of things. If someone was operating in a community where they didn't have access to a strong, active collective or another network of doulas that, you know I don't want to say mentor, but you know what it's like when you have someone who's been in the business forever, I feel like that would really dampen their experience just because of how much I've gained from my experience with this sort of awesome end of the spectrum. (Participant 8)*

Participants described how connections to social service organizations facilitated effective and sustainable abortion doula support. They valued being able to make direct referrals to women's centres, emergency or transitional housing, and sexual assault centres. In turn, these organizations would refer clients to abortion doulas as needed. Some participants aimed to integrate abortion doulas into these organizations to make support available in-house.

Abortion doulas also described positive and collaborative relationships with abortion providers and local clinics improved experiences for both doulas and clients. Some clinics would directly refer clients perceived as needing extra support. One abortion doula described how providers had contributed to their doula collective's training sessions, including an invitation to tour the clinic and meet providers:

*I feel that's been the most valuable connection out of everything, for me to have like direct knowledge of medicines and exactly what the procedure looks like and what it's going to look like for the client before, during and after. That has all come from our hospital and the connections that our collective has made for the group of doulas that are a part of it. The real-life training from the actual people doing the procedure really, I feel, arms me with more knowledge to be able to support my clients when they are in that specific space. (Participant 8)*

## Theme 6: Beyond the individual

Lastly, abortion doulas spoke about their roles beyond the individual client-doula relationship. They saw their work existing in a larger continuum of reproductive rights and justice work. Most participants described engaging in advocacy, either as an abortion doula or as a member of a local, regional, and/or national abortion rights organization/network. They advocated for expanded access to abortion in several ways, describing efforts to 1) encourage more prescribers to prescribe medication abortion; 2) end belief-based denials of care; and 3) ban the use of graphic anti-abortion billboards and flyers. Abortion doulas also felt a responsibility to normalize and de-stigmatize abortion. They described on-going conversations with clinicians, personal and professional networks, family and friends, and other doulas:

> *No matter what your reason for choosing to terminate your pregnancy, it's valid. So I think continuing campaigns like that are focused on removing taboos, that will help us here. And for us to do some of the legwork as well by speaking about it openly to those around us. (Participant 7)*

## Discussion

To our knowledge, this is the first published study in Canada to examine abortion doulas' experience, training, and professional needs. We found abortion doulas played a supportive role in both inter- and intra-systems navigation, including direct provision of transportation, sharing of information, and securing funds for out-of-pocket costs. They provided physical and emotional support in the form of continuous after-care. They also played a social role as advocates for de-stigmatization of abortion and improvements to access.

Despite decriminalization and public funding, the pathway to accessing abortion in Canada requires patients frequently face logistical complexities, travel burdens and private expenditures. The care environment may be unapproachable, and clinic policies encumbering. These barriers to seeking abortion care are acute for those who also face intersecting barriers to equitable health care more broadly: youth, people with disabilities, people living rural areas, people experiencing housing precarity or poverty, Indigenous people and people of colour, members of 2SLGTBQIA + communities, and people without Medicare coverage. Working within a reproductive justice framework, doulas provide specific and pragmatic solutions to de-link and overcome these barriers. Abortion doulas have the potential to interrupt structural barriers to care, affirming client rights to have a child or not have a child under the conditions of one's choosing [57]. The challenges doulas encountered included what they described as a sometimes surprising complexity in their clients' needs, and persistent dilemmas in sustainable remuneration whether practicing independently or within a doula organization/network. Facilitators to their practice included comprehensive training with in-depth clinical content, and strong partnerships with community organizations, abortion care providers, and facilities.

The dominance of transportation support as a key element of doula service is notable. It was beyond our study to articulate, but we recognize there may be cost, safety and/or liability considerations for abortion doulas driving clients to appointments. Despite the implementation of medication abortion in primary care [60] abortion seekers continue to face substantial travel burdens. Inter- and intra-provincial bus or train services are extremely limited in some parts of Canada. Even within a city, abortion doulas may be driving patients home from procedural abortions where escorted drives are required by clinic policy. Doulas expressed frustration that volunteers and/or tenuously funded community organizations were left to provide logistical and financial support needs left unmet by the formal healthcare system.

The informational support role described by abortion doulas differs significantly from the common manifestation of information support in the perinatal contexts. While perinatal doulas may be relied upon for some basic prenatal information and supplemental referrals, such as to lactation consultants or pelvic floor physiotherapists, they are generally not expected to provide information about where to seek a primary pregnancy care provider. By contrast, abortion doulas often act as 'navigators'; arranging first contact for their clients and recommending from whom and how abortion care can be sought. Abortion care pathways remain stigmatized and shrouded in secrecy, particularly in the second and third trimesters. Abortion doulas assume a heavy burden to prepare and share an in-depth understanding of the inner workings of formal healthcare systems, while working largely outside of those systems as unregulated, nonclinical professionals.

Although abortion is very safe, and most abortions in Canada occur in the first trimester [8], abortion doulas described confronting surprising clinical complexity and did not feel adequately prepared. Indeed, our research team did not expect these complex cases to play such a large part in client base of those seeking doula support; we anticipated a doula role largely limited to the accompaniment of clients through a process of medication abortions at home. Abortion doulas characterized most of their clients as experiencing intersecting forms of social oppression, which contributed to clinical complexity. For example, youth or newcomers to Canada unaware of their rights and options who consequently connected with care at later gestational duration, when care became more complex and required more extensive travel and logistical support. Conceptualizing the work of abortion doulas within a framework of reproductive justice helps to move beyond binary understandings of 'choice' and 'access'. While pathways to accessing abortion may be straight-forward for many people seeking care in Canada, the experiences of the doulas we interviewed exposes unmet needs for support experienced by clients facing intersectional structural inequities.

Although completely decriminalized and largely publicly funded, abortion remains misunderstood, with particularly acute consequences for abortion-seekers experiencing intersecting oppressions. While many patients would feel comfortable approaching a physician or nurse practitioner for information about abortion [61], participating abortion doulas reported their clients turned to them for detailed clinical information. They recommended this information be included in abortion doula training. Doulas reported fielding questions including: which medications are used; their pharmacological mechanisms of action; what instruments are used during a procedural abortion, and for what purpose; what clients could expect in terms of symptoms, side effects, and recovery time. While participants described the importance of this knowledge to facilitate their practice, efforts to prepare doulas to answer these questions may raise concerns about the potential for practicing outside of the non-clinical doula scope [33].

To practice, doulas do not require certification, licensing, or registration. They do not have regulators to which concerns about practice could be reported. The client needs described by doulas in this study raise the question of to what extent these non-clinical professionals could or should be relied upon to fill gaps in information and clinical support that the health system fails to provide, i.e., for abortion seekers facing intersectional barriers to accessible care. Abortion doulas may also face challenges obtaining liability insurance, thus assuming personal risks.

With few exceptions [62], there is little public funding for the doula role in Canada [63] whether in the realm of birth support or abortion. While the doula profession as a whole faces challenges to remuneration [64–66], participants in this study described this as particularly fraught in the context of abortion care. While future parents may be willing to pay privately for doula services in the context of birth preparation and the anticipated positive experience of having a newborn, abortion doulas felt uncomfortable and guilty asking for money from

clients seeking pregnancy termination. Furthermore, some of the participants in this study reported paying for client needs out of their own pockets, effectively personally subsidizing gaps in financial support in the formal health system.

Finally, abortion doulas described partnerships with clinics as facilitative, however, they emphasized they must remain independent, and that this may even involve supporting clients in complaints regarding clinician misconduct. Clinic policies and staff attitudes towards doulas are an area for future research. Similarly, there is a need for Canadian research on the perspectives of clients who have received abortion doula support.

## Limitations

This study has several limitations. The participants were recruited from a large list of doulas we generated using publicly available online information. Doulas without an online profile, or who we failed to include, or who did not publicly identify as supporting abortion-seekers, were excluded. Abortion doulas who choose not to publicly identify as such due to fear of harassment or other consequences may have unique perspectives we were unable to consider.

## Conclusion

The landscape of abortion access in Canada has changed dramatically in recent years with the introduction of mifepristone for medication abortion, and COVID-19-driven changes to tele-medicine. Despite improvements to access in Canada, barriers remain, particularly for under-served groups experiencing intersecting forms of social oppression. Abortion doulas may be particularly well-suited to mitigate barriers. Our qualitative study among abortion doulas identified that their role includes addressing client needs for information, transportation, and support navigating the healthcare system. They encounter clients with surprisingly complex social and clinical needs, and report inadequate training about clinical aspects of abortion, and little or no remuneration. In addition to comprehensive training, they report peer networks, partnerships with social service organizations and with abortion clinics facilitate their doula practice. Future research should investigate the attitudes, policies and procedures regarding doulas among abortion providers and clinics in Canada and the experiences of abortion doula clients.

## Supporting Information

**S1 File. Human Participants Research Checklist.**
(DOCX)

## Author contributions

**Conceptualization:** Martha Paynter.

**Data curation:** Martha Paynter, Clare Heggie, Anja McLeod.

**Formal analysis:** Martha Paynter, Clare Heggie, Anja McLeod, Mélina Castonguay, Melissa Fuller.

**Funding acquisition:** Wendy V. Norman.

**Methodology:** Martha Paynter, Anja McLeod, Mélina Castonguay, Melissa Fuller.

**Project administration:** Clare Heggie, Anja McLeod.

**Supervision:** Martha Paynter, Mélina Castonguay, Melissa Fuller, Wendy V. Norman.

**Validation:** Mélina Castonguay, Melissa Fuller.

**Writing – original draft:** Martha Paynter, Clare Heggie.

**Writing – review & editing:** Martha Paynter, Mélina Castonguay, Melissa Fuller, Wendy V. Norman.

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
