## [Decision Letter · Decision Letter 0]

3 Jul 2024

PONE-D-24-15526The role of doulas in abortion care in Canada: A qualitative studyPLOS ONE

Dear Dr. Paynter,

Thank you for submitting your manuscript to PLOS ONE. After careful consideration, we feel that it has merit but does not fully meet PLOS ONE’s publication criteria as it currently stands. Therefore, we invite you to submit a revised version of the manuscript that addresses the points raised during the review process. Both reviewers raised important points that will improve the reporting of the methods and the findings of this study. I would encourage you and your team to incorporate the suggestions for a theoretical grounding and deeper exploration of some of the themes proposed by Reviewer 1. In addition, the requested additions to the methods section by Reviewer 2 highlight a similar need for both clarity and depth, but with regards to the methods.

We look forward to receiving your revised manuscript.

Kind regards,

Andrea Knittel

Academic Editor

PLOS ONE

Reviewers' comments:

Reviewer's Responses to Questions

**Comments to the Author**

1. Is the manuscript technically sound, and do the data support the conclusions?

Reviewer #1: Yes

Reviewer #2: Yes

2. Has the statistical analysis been performed appropriately and rigorously? 

Reviewer #1: N/A

Reviewer #2: N/A

3. Have the authors made all data underlying the findings in their manuscript fully available?

Reviewer #1: No

Reviewer #2: Yes

4. Is the manuscript presented in an intelligible fashion and written in standard English?

Reviewer #1: Yes

Reviewer #2: Yes

5. Review Comments to the Author

Reviewer #1: Thanks so much for the opportunity to review this paper. I really enjoyed reading it and I think it is an incredibly important topic. While I don’t work in the Canadian context, I think the findings here are very applicable to the US, perhaps especially because of the challenging legal framework in the US. It’s well written and the flow makes logical sense. In sum, I really like this paper and I think it should be published.

I do have a couple thoughts about potential revisions which could be either quite minor (a shift in description) or a bit more substantial, depending on the preferences of the authors and editor. While I loved the content of the paper, I felt a little underwhelmed by the framing. The three themes (what do abortion doulas do; challenges to abortion doula work; facilitators to abortion doula work), just didn’t feel like themes that emerged from the data. These felt like they could have easily come from the aims of the project. I am not questioning organizing the data this way — It makes logical sense! I just would have rather it was noted that the paper was organized along these lines and not that these emerged through analysis. This feels very nit-picky, but the framing of these categories as themes just made me feel that while qualitative methods were used, perhaps qualitative methodology was not.

Similarly, I just felt a bit underwhelmed by the sub themes. Complexity, for example, felt like there was an under interrogation of the themes. Both the stigmatization of people who engage in sex work, substances, etc and preparedness for second and third trimester abortions felt like gigantic topics and I wanted a better understanding of how and why you connected them. Both being “complex” just felt under developed. I would say in general I felt like the paper could make more of a theoretical intervention - or at least draw on theory — more than it does. For example, there were many references to stigma but no mention of Goffman or the many, many advances on Goffman’s theory in the reproductive health space. I was very surprised not to see mention of Reproductive Justice. While race and racism was mentioned briefly in the background, it wasn’t at all present in the paper. Was this not present in the data? There was a mention of birth doulas as an intervention on racism — how is this considered in abortion doula work? (Again, this feels like it relates back to wanting more about if and how doulas intervene upon interpersonal and structural stigma). I also thought Sustainability could have gone a bit further — the financial instability is such a huge challenge. I wanted to know more. The idea of putting a band aid on the problems — I just wanted to know more about that kind of meaning making, how the doulas saw their role and what it meant for them; how does this intersect with advocacy work? Are the doulas dealing with burn out from this?

I’m really torn in my feedback. I don’t want to ask the authors to write a paper that isn’t what they want to write. If the goal is for this to be a descriptive paper, I think it is meeting that goal and perhaps shifting the headings a bit (I wouldn’t call them themes) and highlighting the potential implications of the work that could be explored further would help address my concerns. What I would love to see is an expansion of the themes — say more than “complex” and “community” — and a digging into the meaning of these barriers and facilitators. I think it would make the paper more powerful and really has potential to expand the literature on doula work, reproductive justice, abortion, and stigma. But, if the goal is a more straightforward descriptive paper, I think that is fine too as long as that is clear to the reader.

Reviewer #2: This is a strong manuscript reporting on the results of a qualitative study with 15 doulas who provide abortion care in Canada. The authors do a commendable job in the introduction laying out the context of accessing abortion care in Canada and sensitively describing the many barriers that people face and that may be rendered more complex for people belonging to a wide range of underserved groups. The results are compellingly presented and are supported by the data, as are the conclusions.

My minor comments focus on the methods section. Given the thoroughness the authors provide in other sections, this section is a bit sparse. Specifically, it would be helpful for situating the manuscript to have more details about the following:

1. The authors note that they conducted the study in partnership with non-academic experts in doula care, which is a strength. Can they provide more details about how this partnership came about? Did it exist prior to the study?

2. As noted, the authors provide a sensitive review of the various underserved groups who may face additional complexities when accessing abortion care. Did any members of the study team (including the non-academic partners) have lived expertise of any of the conditions or identities experienced by these groups?

3. Roughly one fifth of the doulas recruited to participate in interviews agreed to do so, which is an adequate sample. However, it would be helpful to hear more about the recruitment efforts: what role, if any, did the non-academic experts play in designing the recruitment materials? How did the team attempt to make study participation appealing (beyond the honorarium, and including what information was provided in the recruitment message regarding confidentiality and assuring participant safety)?

4. It’s noted that the interviews were conducted on Zoom; how many interviewers were on each call? Did the non-academic partners co-facilitate any of the interviews?

6. PLOS authors have the option to publish the peer review history of their article (what does this mean? ). If published, this will include your full peer review and any attached files.

**Do you want your identity to be public for this peer review?** For information about this choice, including consent withdrawal, please see our Privacy Policy .

Reviewer #1: No

Reviewer #2: No

---

## [Author Response · Author response to Decision Letter 0]

19 Sep 2024

Responses to Reviewers

Reviewer #1:

While I loved the content of the paper, I felt a little underwhelmed by the framing. The three themes (what do abortion doulas do; challenges to abortion doula work; facilitators to abortion doula work), just didn’t feel like themes that emerged from the data. These felt like they could have easily come from the aims of the project. I just would have rather it was noted that the paper was organized along these lines and not that these emerged through analysis. Thank you for your thoughtful feedback. We have revisited our analysis and re-organized into six key themes: Inter and intra-systems navigation; After-care; In over their heads; Balancing act; Support for the supporter; and Beyond the individual. We feel that these themes speak more accurately to the work of abortion doulas represented in the data.

I just felt a bit underwhelmed by the sub themes See comment above.

… I would say in general I felt like the paper could make more of a theoretical intervention - or at least draw on theory — more than it does. Thank you, we have integrated reproductive justice theory into the framing and discussion of key themes.

For example, there were many references to stigma but no mention of Goffman or the many, many advances on Goffman’s theory in the reproductive health space. We have added further background on abortion stigma on lines 106-113.

I was very surprised not to see mention of Reproductive Justice. While race and racism was mentioned briefly in the background, it wasn’t at all present in the paper. Was this not present in the data? Thank you, we do work from a framework of reproductive justice, but this was not made explicit in the paper. We have added a section in the methods on how reproductive justice theory guided our study design and implementation. While doulas did not explicitly mention reproductive justice, we have commented throughout on where their work aligns with a vision of reproductive justice.

There was a mention of birth doulas as an intervention on racism — how is this considered in abortion doula work? (Again, this feels like it relates back to wanting more about if and how doulas intervene upon interpersonal and structural stigma) We have added citations to explain that, like in perinatal care, racialized and queer patients may face structural barriers to abortion care.

I also thought Sustainability could have gone a bit further — the financial instability is such a huge challenge. I wanted to know more. The idea of putting a band aid on the problems — I just wanted to know more about that kind of meaning making, how the doulas saw their role and what it meant for them; how does this intersect with advocacy work? Are the doulas dealing with burn out from this? We have collapsed sustainability into Theme 4: A balancing act. We feel this better communicates how the doulas saw their role and expressed the challenges regarding sustainability, burn-out, and the ethical dilemmas between providing equity-based care and seeking renumeration. Within this new theme, we have expanded on both the financial instability and the doulas perceptions of short-term fixes to structural inequalities in care.

What I would love to see is an expansion of the themes — say more than “complex” and “community” — and a digging into the meaning of these barriers and facilitators. I think it would make the paper more powerful and really has potential to expand the literature on doula work, reproductive justice, abortion, and stigma Thank you, see comment above re: expansion of themes. Additionally, we collapsed several themes that were previously presented as either “barriers” or “facilitators” to more accurately reflect the nuanced nature of the abortion doula work, where healthcare systems and models of work can both facilitate and challenge the work.

Reviewer #2:

The authors note that they conducted the study in partnership with non-academic experts in doula care, which is a strength. Can they provide more details about how this partnership came about? Did it exist prior to the study? We have added further detail on lines 161-167.

As noted, the authors provide a sensitive review of the various underserved groups who may face additional complexities when accessing abortion care. Did any members of the study team (including the non-academic partners) have lived expertise of any of the conditions or identities experienced by these groups? Added detail on lines 168-9.

Roughly one fifth of the doulas recruited to participate in interviews agreed to do so, which is an adequate sample. However, it would be helpful to hear more about the recruitment efforts: what role, if any, did the non-academic experts play in designing the recruitment materials? How did the team attempt to make study participation appealing (beyond the honorarium, and including what information was provided in the recruitment message regarding confidentiality and assuring participant safety)?

Added detail on line 175-178.

It’s noted that the interviews were conducted on Zoom; how many interviewers were on each call? Did the non-academic partners co-facilitate any of the interviews?

Added detail on line 195-197.

---

## [Editor Report · Decision Letter 1]

4 Nov 2024

The role of doulas in abortion care in Canada: A qualitative study

PONE-D-24-15526R1

Dear Dr. Paynter,

We’re pleased to inform you that your manuscript has been judged scientifically suitable for publication and will be formally accepted for publication once it meets all outstanding technical requirements.

Kind regards,

Andrea K. Knittel

Academic Editor

PLOS ONE

Additional Editor Comments (optional):

There are a few very small typographical/style issues to resolve prior to publication. These can be resolved with any outstanding technical comments.

1. Please write out the meaning of the 2SLGBTQIA+ acronym the first time it is used (line 91).

2. Some of the references were duplicated at the end of the sentence in line 168.

3. Add back in the name of the institution when referencing the name of the IRB (lines 205-206).

4. Please write out the meaning of the HCP acronym the first time it is used (line 299).

5. The portion of the sentence in line 577 should read "during a procedural abortion, and for what purpose;" instead of the semicolon currently in the middle of the phrase.

---

## [Editor Report · Acceptance letter]

PONE-D-24-15526R1

PLOS ONE

Dear Dr. Paynter,

I'm pleased to inform you that your manuscript has been deemed suitable for publication in PLOS ONE. Congratulations! Your manuscript is now being handed over to our production team.

Kind regards,

on behalf of

Dr. Andrea K. Knittel

Academic Editor

PLOS ONE